# Platypnea-Orthodeoxia Syndrome Manifesting as an Early Complication after Lower Bilobectomy

Carmelina C. Zirafa [1,*,†] , Alessandra Lenzini [2,†] , Paolo Spontoni [3] , Claudia Cariello [4] , Luca Doroni [4] , Adrea Pieroni [3] , Anna S. Petronio [3] and Franca Melfi [1]

1   Minimally Invasive and Robotic Thoracic Surgery Division, Robotic Multispecialty Center of Surgery, University Hospital of Pisa, 56124 Pisa, Italy
2   Thoracic Surgery Unit, Department of Surgical, Medical, Molecular, Pathology and Critical Care, University Hospital of Pisa, 56124 Pisa, Italy
3   Cardiac Catheterization Laboratory, Cardiothoracic and Vascular Department, University Hospital of Pisa, 56124 Pisa, Italy
4   Cardiothoracic and Vascular Anaesthesia and Intensive Care Unit, Department of Anesthesia and Critical Care Medicine, University Hospital of Pisa, 56124 Pisa, Italy
*   Correspondence: c.zirafa@gmail.com; Tel.: +39-05-0995-814
†   These authors contributed equally to this work.

**Abstract:** Platypnea-orthodeoxia syndrome (POS) is an uncommon clinical condition characterized by orthostatic dyspnea and hypoxemia. The case of a female patient who manifested postoperative episodes of sudden oxygen desaturation, dyspnea, and systemic arterial hypotension following lower bilobectomy for lung adenocarcinoma was reported. After meticulous clinical investigations, the patient proved to be affected by a rare form of postural dyspnea: platypnea-orthodeoxia syndrome, a clinical disorder described in the middle of the last century. The pathophysiology was found in an intracardiac mechanism of right-to-left blood shunt, combined with lung and chest wall modification. Atrial septal defect, such as patent foramen ovale (PFO), is a common cause of platypnea-orthodeoxia syndrome; the rescue closure of PFO usually allows for an immediate and consistent improvement of the symptoms.

**Keywords:** platypnea-orthodeoxia; bilobectomy; thoracic surgery; lung cancer; lung resection; oxygen desaturation; orthostatic hypotension





## 1. Introduction

Platypnea-orthodeoxia syndrome is a rare clinical condition characterized by dyspnea and arterial desaturation, occurring in orthostatism and vanishing in the supine position.

First reported in a case of a post-traumatic intrathoracic arteriovenous shunt in 1949 [1] and described for the first time in 1984 [2], POS appears to be a complex condition, derived from one of the following mechanisms: intracardiac shunt, pulmonary vascular shunt, ventilation/perfusion mismatch, or a combination of them. The pathogenetic mechanisms involve both functional and anatomical components: this condition is associated with cardiac, pulmonary, abdominal, and vascular disease combined with a vascular shunt. The presence of pulmonary hypertension in conjunction with a patent foramen ovale (PFO) or an atrial septal defect is the most accepted cause. The upright position is the trigger able to produce conformational changes of interatrial communication, increasing the proportion of blood flow from the inferior vena cava through the defect [3].

The crucial event in POS pathophysiology is the intermixing of deoxygenated venous blood with arterial blood. In the case of PFO or atrial septal defect, deoxygenated blood is shunted from the right to the left atrium through an atrial septal defect, and the right atrium pressure could be normal or increased. Otherwise, the intrapulmonary mechanism is less common and it may be associated with vascular abnormalities, capillary lung dilatation, ventilation/perfusion mismatch, and a wide range of parenchymal diseases [4].

In physiological conditions, minimal right-to-left shunts are ordinary to observe in healthy subjects, without any hemodynamic consequences. In pathological conditions, venous deoxygenated blood flows directly into the systemic arterial flow: the amount of the shunt determines the severity of hypoxemia and other symptoms. In the case of POS, the shunt fraction required to reduce arterial oxygen pressure below 70 mmHg is approximately 20–25%, while a shunt fraction of 50% leads to severe hypoxemia ($PaO_2$ < 40 mmHg) [5].

Regarding the diagnostic approach if POS is suspected, the intracardiac mechanism is the first that must be investigated, considering its higher prevalence compared with the pulmonary cause. The diagnosis of the intracardiac shunt is most readily established by transthoracic echocardiogram, improved by using agitated saline contrast during the Valsalva maneuver [6]. Transesophageal echocardiography is recommended, if the diagnostic suspicion remains after the transthoracic exam, to confirm the interatrial septum as well as to plan the potential closure strategy of the defect.

## 2. Case Presentation

A 76-year-old female patient was admitted to our surgical department in January 2021. She was a current smoker (45 pack-year), affected by an atrial septal aneurysm (ASA), obstructive pulmonary disease (COPD), mixed anxiety-depressive disorder (MADD), gastroesophageal reflux disease (GERD), osteoporosis. Furthermore, she had a history of cerebral vascular disease with a previous transient ischemic attack in 2010. Her previous surgeries were a cholecystectomy, a hernioplasty, and a benign breast nodule excision. Her home therapy included escitalopram, lysine acetylsalicylate, esomeprazole, and cholecalciferol. She was referred to our unit for the management of a large lung mass, revealed by a routine chest X-ray. The preoperative computed tomography (CT) scan (Figure 1) showed a lesion in the right lower lobe ($69 \times 54 \times 76$ cm) with suspected infiltration of costal pleura and distal occlusion of the bronchus intermedius associated with hilar lymphadenopathy. Therefore, positron emission tomography (PET) CT scan detected increased glucose metabolism exclusively in the lung mass (SUV max 24.08) and in the right hilar lymphadenopathy (SUV max 22.48). The bronchoscopy image showed endobronchial vegetation occluding the apical segment of the right lower lobar bronchus. Histological examination of bronchial biopsy revealed a lung adenocarcinoma wild-type with a PD-L1 expression of 30%. Last, a cranial CT scan was performed and it was negative for brain metastasis. The clinical staging of the tumor resulted in cT4N1M0. The patient performed global spirometry, which revealed a mild obstructive ventilation defect, with a forced expiratory volume in one second (FEV1) of 1.70 L (105%) and a Tiffeneau index of 0.64. The cardiological evaluation did not detect pathological conditions, and the electrocardiogram showed sinus rhythm and regular QT interval. The invasive mediastinal staging was performed through EBUS-TBNA, which did not reveal any mediastinal positive lymphadenopathy.

After the multidisciplinary oncology tumor board reviewed the case, the patient became a candidate for lung surgery. Extrapleural lower bilobectomy with the removal of the VI rib was performed via posterolateral thoracotomy. The surgery lasted 205 min with no intraoperative complications; two chest tubes were positioned at the end of the operation. Pathological analysis diagnosed a "poorly differentiated adenocarcinoma with a solid pattern, infiltrating the parietal pleural, with widespread neoplastic embolization, pT4N1G3". After the surgical procedure, the patient was admitted to our sub-intensive care unit for multiparameter monitoring. The patient required a blood transfusion on the day of surgery. The postoperative bedside chest X-ray showed incomplete lung expansion (Figure 2).

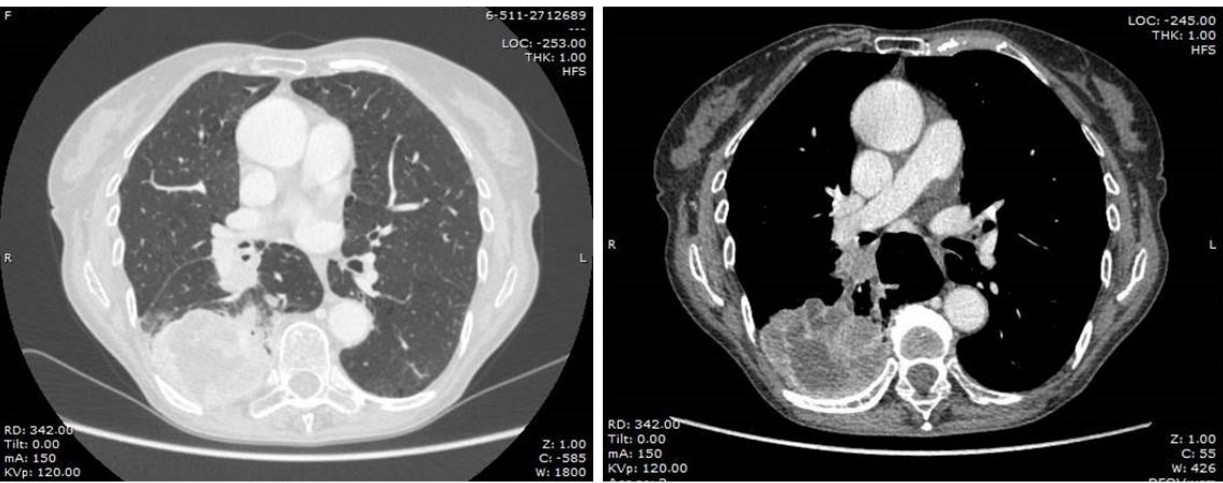

**Figure 1.** Pre-operative chest CT-scan.

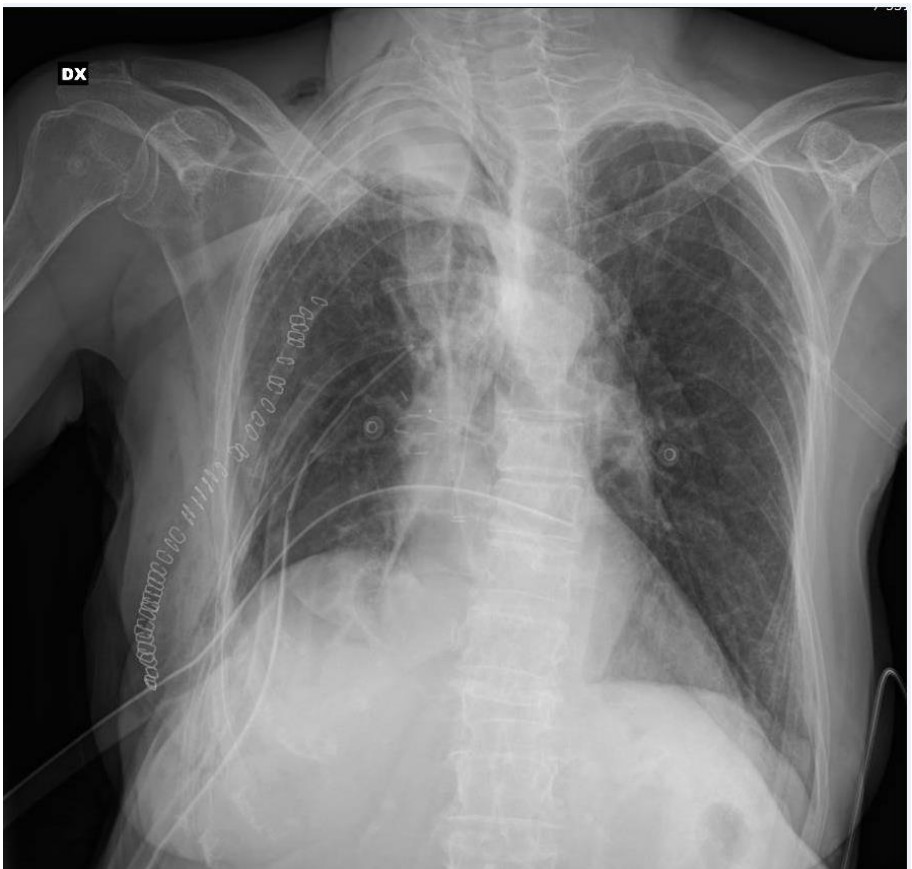

**Figure 2.** Post-operative chest X-ray.

On the first postoperative day, after mobilization, the patient presented a sudden oxygen desaturation episode with peripheral capillary oxygen saturation ($SpO_2$) of 80% and arterial hypotension (85/55 mmHg) associated with a soporose state. An arterial blood gas (ABD) revealed a partial pressure of oxygen ($PaO_2$) of 60 mmHg. A Ventury mask was used to maintain adequate oxygenation and a toilet bronchoscopy was readily performed to aspirate retained secretions.

The neurological examination revealed the absence of bulbomimic reflex, anisocoria, and positive Mingazzini maneuvers. A brain CT scan revealed two small thalamic areas attributable to suspicious ischemic lesions; therefore, the neurologist recom-

mended rest in a supine position and administration of acetylsalicylic acid in addition to low-molecular-weight heparin. The transthoracic echocardiogram (TTE) confirmed the presence of ASA without shunt signs.

After 24 h, the patient was awake and active, staying in a supine position. A brain CT scan was repeated and showed the stability of the ischemic lesions, compared with the previous scan.

The soporose condition was alternated with an awake state in the following days, having to keep the flow rate of oxygen via nasal cannula at about 6–8 L/min to maintain $SpO_2 > 90\%$.

On the third postoperative day, an acute neurologic deficit with left-sided paralysis, associated with desaturation and hypotension, occurred during a new attempt to mobilize the patient, with a complete resolution of symptoms in the supine position.

Considering the onset of these neurological symptoms, the patient underwent a new brain CT scan, which did not reveal modifications of the thalamic lesions. According to the neurologist's recommendations, the patient maintained the supine position in the following days. A new similar episode of desaturation ($SpO_2$ of 70%) with severe hypotension was observed on the seventh postoperative day during another attempt at the patient's mobilization.

The CT scan of the thorax was performed to investigate pulmonary embolism or other parenchymal alterations that could explain the symptoms, excluding possible pulmonary pathogenesis. Furthermore, transthoracic echocardiography showed EF (ejection fraction) of 60%, PAPs (pulmonary arterial systolic pressure) of 30 mmHg, and atrial septal aneurysm 3RL. At that time, platypnea-orthodeoxia syndrome was suspected. Then, the clinical condition progressively worsened, so the patient was moved to an intensive care unit to treat respiratory failure with noninvasive mechanical ventilation. The patient was transferred to the surgical ward after four days when an improvement in the clinical condition was obtained.

To confirm the suspicion of platypnea-orthodeoxia syndrome, transoesophageal echocardiography was performed, which showed an interatrial septum with an exuberant hyperdynamic movement from the right to the left atrium, compatible with an important patent foramen ovale (PFO) (Videos S1 and S2). In addition, microembolic signals in the basal cerebral arteries were observed with contrast-enhanced transcranial Doppler ultrasound.

The patient was referred to the Cardiac Catheterization Laboratory for percutaneous closure of PFO via a transesophageal echocardiogram-guided Amplatzer Multi-Fenestrated Septal Occluder (Video S3). The patient was able to stand up with no symptoms the day after implantation, and she was discharged on the third postimplantation day. All the details concerning the hospital stay are reported in Table 1. One-month follow-up examination showed a good and stable condition (Figure 3). With regard to the oncological follow-up, the histological examination led to the diagnosis of invasive adenocarcinoma with predominant solid pattern pT4N1 G3; the comprehensive molecular profiling did not reveal any mutation. According to the pathological stage, the patient was admittable to adjuvant chemotherapy; the performance status after surgery did not allow the administration of systemic therapy. The patient underwent a total body CT scan six months after surgery which detected local cancer progression and multiple bone metastases.

**Table 1.** Patient's timeline.

| Recovery Day | Signs and Symptoms | Blood Tests | | Instrumental Investigation | Treatment |
|---|---|---|---|---|---|
| Day -1 | Good clinical conditions | CBC | Hb: 12.5 g/dL<br>WBC: 9.16 $\times$ $10^3$ µL | ECG: sinus rhythm<br>Echocardiography: ASA without a shunt. EF 60%<br>SARS-CoV test: negative | |
| Day 0<br>SURGERY | SpO$_2$: 90–100%<br>BP: 110/90 mmHg<br>HR: 70 bpm | CBC<br>ABG | Hb 9.5 g/dL<br>WBC 13.61 $\times$ $10^3$ µL<br>pH 7.41<br>pO$_2$ 68 mmHg<br>pCO$_2$ 42<br>SO$_2$ 96% | Chest X-ray: incomplete right lung expansion | Single-unit blood transfusion<br>Antibiotic therapy<br>Oxygen therapy (nasal cannula, 4 L/min) |
| Day 1 | In the morning episode of sudden desaturation (SpO$_2$ 80%)<br>Soporose state<br>Anisocoria<br>SpO$_2$: 100%<br>BP: 90/55 mmHg<br>HR: 70 bpm | CBC<br>ABG | Hb: 9.8 g/dL<br>WBC: 12.03 $\times$ $10^3$ µL: pH 7.4<br>pO$_2$ 50 mmHg<br>pCO$_2$ 41 mmHg<br>SO$_2$ 88% | TTE: ASA without a shunt<br>Brain CT scan: two small thalamic areas attributable to suspicious ischemic lesions<br>EEG: stage N1–N2 sleep | Tracheobronchial toilet<br>Corticosteroids<br>Oxygen therapy (Venturi mask 15 L) |
| Day 2 | Awake and active<br>SpO$_2$ 100%<br>BP: 100/60 mmHg<br>HR: 60 bpm | CBC<br>ABG | Hb: 8.7 g/dL<br>WBC: 9.06 $\times$ $10^3$ µL<br>pH 7.4<br>pO$_2$ 78 mmHg<br>pCO$_2$ 43 mmHg<br>SO$_2$ 98% | Chest X-ray: incomplete right lung expansion<br>Brain CT scan: stable | Single-unit blood transfusion<br>Neuro-rehabilitation exercises |
| Day 3 | Acute neurologic deficit with left-sided paralysis<br>SpO$_2$ 100%<br>BP: 120/70 mmHg<br>HR: 60 bpm | CBC | Hb: 9.7 g/dL<br>WBC: 7.71 $\times$ $10^3$ µL | Chest X-ray: stable<br>CT neck\brain angiography: stable | Supine position<br>Acetylsalicylic acid 100 mg<br>Atorvastatin 20 mg |
| Day 4 | Awake and stable<br>SpO$_2$ 100%<br>BP: 130/90 mmHg<br>HR: 80 bpm<br>An episode of autonomous mobilization, with SpO$_2$ 84% | CBC<br>ABG | Hb: 10.1 g/dL<br>WBC: 6.51 $\times$ $10^3$ µL<br>pH 7.55<br>pO$_2$ 41 mmHg<br>pCO$_2$ 37 mmHg<br>sO$_2$ 85% | Chest X-ray stable | Supine position<br>Oxygen therapy (Venturi mask 15 L) |
| Day 5 | Awake and stable<br>SpO$_2$ 100%<br>BP: 120/70 mmHg<br>HR: 60 bpm | CBC | Hb: 11 g/dL<br>WBC: 5.61 $\times$ $10^3$ µL<br>CRP 2.76 mg/dL | | Supine position<br>Removal of basal chest drainage |

**Table 1.** *Cont.*

| Recovery Day | Signs and Symptoms | Blood Tests | | Instrumental Investigation | Treatment |
|---|---|---|---|---|---|
| Day 6 | Awake and stable<br>SpO$_2$ 90–100%<br>BP: 120/70 mmHg<br>HR: 70 bpm | | | Brain CT scan: stable | Partial mobilization |
| Day 7 | Postural dyspnea<br>SpO$_2$ 80%<br>BP: 120/80 mmHg<br>HR: 110 bpm | ABG 9 am<br><br>ABG 4 pm | Hb: 10.9 g/dL<br>WBC: 7.85 × 10$^3$ μL<br>CRP: 1.63 mg/dL<br>pH 7.53<br>pO$_2$ 37 mmHg<br>pCO$_2$ 38 mmHg<br>Lac 2.0 mmol/L<br>SO$_2$: 78%<br>Ph 7.61<br>pO$_2$ 37 mmHg<br>pCO$_2$ 27 mmHg<br>SO$_2$ 80% | Chest X-ray: stable<br>CT chest angiography: no signs of pulmonary embolism<br>ECG: sinus rhythm<br>TTE: EF 60%, ASA<br>COVID-19 test negative | Clopidogrel 75 mg<br>Oxygen therapy (Venturi mask 15 L) |
| Days 8–11<br>Intensive care unit | | ABG | pH 7.46<br>pO$_2$ 126 mmHg<br>pCO$_2$ 41 mmHg<br>SO$_2$ 97.5% | Intensive monitoring | Noninvasive ventilation<br>Pression support 14 cm H$_2$O<br>Peep3 cm H$_2$O<br>FiO$_2$ 35%<br>Sildenafil 25 mg<br>Blood transfusion |
| Days 12–15 | Awake and stable<br>SpO$_2$: 91–100%<br>BP: 110/60 mmHg<br>HR: 60–70 bpm | CBC<br><br>ABG | Hb: 11.3 g/dL<br>WBC: 7.64 × 10$^3$ μL<br>pH 7.45<br>pO$_2$ 63 mmHg<br>pCO$_2$ 40 mmHg<br>SO$_2$ 93.3% | Chest X-ray: complete right lung expansion<br>COVID-19 test: negative | Oxygen therapy (nasal cannula, 3–5 L/min)<br>Removal of apical chest drainage |
| Day 16 | Awake and active<br>SpO$_2$: 85–100%<br>BP: 110/75 mmHg<br>HR: 70 bpm | ABG | pH 7.46<br>pO$_2$ 55 mmHg<br>pCO$_2$ 33 mmHg<br>SO$_2$ 90.6%<br>Lac 0.7 mmol/L | TEE: interatrial septal aneurysm 2 L according to Olivares-Reyes classification. Interatrial communication, attributable to patent foramen ovale (PFO), with the flow of microbubbles in basal portion, after intravenous injection of microbubble contrast agent<br>Enhanced transcranial Doppler ultrasound: microemboli in the basal cerebral arteries, diagnostic for permanent right-to-left shunt. | Oxygen therapy (Venturi mask 15 L) |

**Table 1.** *Cont.*

| Recovery Day | Signs and Symptoms | Blood Tests | | Instrumental Investigation | Treatment |
|---|---|---|---|---|---|
| Day 17 | Awake and active<br>SpO$_2$ 100%<br>BP: 130/70 mmHg<br>HR: 60 bpm | | | | Oxygen therapy (Venturi mask 15 L) |
| Day 18 | Episode of desaturation<br>SpO$_2$ 80%<br>BP 100/70 mmHg<br>HR: 95 bpm | CBC | Hb: 12.5 g/dL<br>WBC: 8.2 × 10$^3$ μL | ECG: sinus rhythm | Bisoprolol 1.25 mg<br>Polygeline 500 mL<br>Oxygen therapy (Venturi mask 15 L) |
| Days 19–23 | Awake and stable | CBC | Hb: 11.5 g/dL<br>WBC: 6.93 × 10$^3$ μL<br>CRP: 1.43 mg/dL | Chest X-rays: stable | Oxygen therapy (nasal cannula, 3 L/min) |
| Day 24 | Percutaneous closure of PFO | CBC | Hb: 9.9 g/dL<br>WBC: 6.48 × 10$^3$ μL | | via transesophageal echocardiogram-guided Amplatzer Multifenestrated Septal Occluder |
| Day 25 | Awake and stable | CBC | Hb: 9.2 g/dL<br>WBC: 6.94 × 10$^3$ μL | TTE: minimal residual shunt | |
| Day 26–27 | Awake and stable in the upright position<br>SpO$_2$ 97% | CBC | Hb: 10 g/dL<br>WBC: 6.9 × 10$^3$ μL | Chest X-ray: stable | Rehabilitation exercises |
| Day 28 | SpO$_2$ 97%<br>BP 112/70 mmHg<br>HR 80 bpm | | | | Hospital discharge |

ABG: arterial blood gas; ASA: Atrial Septum Aneurysm; BP: blood pressure; bpm: beats per minute; CBC: complete blood count; CRP C-reactive protein; CT: Computed Thomography; ECG: electrocardiogram; EEG: electroencephalogram; EF: ejection fraction; FiO$_2$: fraction of inspired O$_2$; Hb: hemoglobin; HR: heart rate; Lac: Lactate; pCO$_2$: partial pressure of carbon dioxide; Peep: positive end-expiratory pressure; PFO: Patent Foramen Ovale; pO$_2$: partial pressure of oxygen; SO$_2$: oxygen saturation; SpO$_2$: peripheral capillary oxygen saturation; TEE: transesophageal echocardiogram; TTE: transthoracic echocardiogram; WBC: white blood cells.

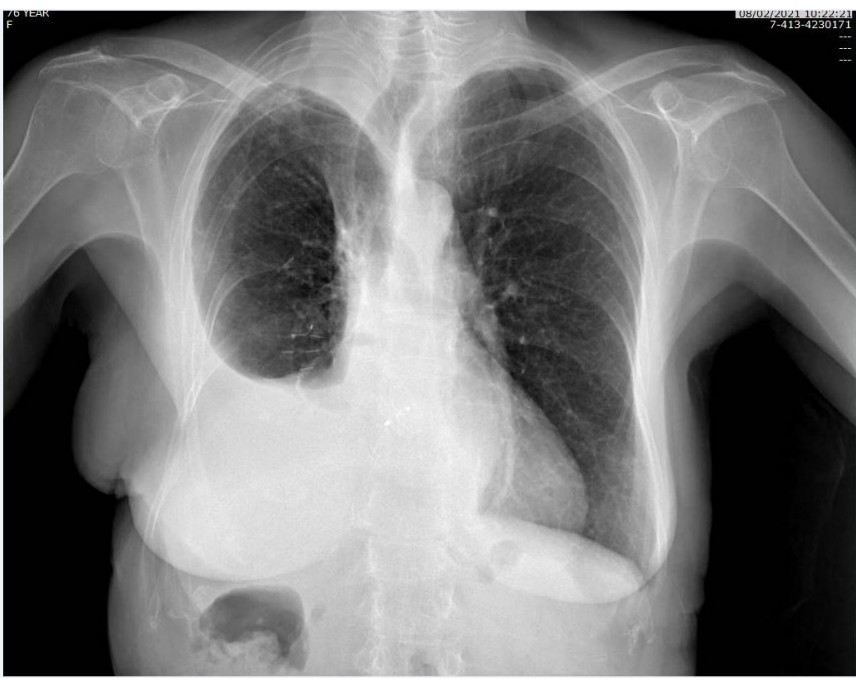

**Figure 3.** Chest X-ray, performed one month after surgery.

The study was conducted according to the guidelines of the Declaration of Helsinki, and approved by the Institutional Review Board of the University Hospital of Pisa (19211/22, 27 July 2022).

### 3. Discussion

Platypnea-orthodeoxia syndrome was identified as a late postoperative complication after pulmonary resection for the first time in 1956 and it was usually reported after pneumonectomy [7–9]. This disease is probably an under-recognized condition, whose pathophysiology remains currently a matter of debate, even if signs and symptoms are paradoxical [10]. Since the past century when the condition was identified for the first time, less than 200 cases of POS have been reported in scientific literature and only a few of them are cases of post-lobectomy POS. The main cause of this rare phenomenon is related to an interatrial right-to-left shunt through a preoperative asymptomatic patent foramen ovale (PFO) or another congenital defect [11]. It is well-established that two distinct conditions are required for the manifestation of the syndrome: anatomical and functional abnormality. Pulmonary resections usually performed for lung cancer (pneumonectomy or lobectomy) can be recognized as anatomical rearrangements, as well as pericardial or aortic pathologies. On the other hand, the presence of a right-to-left shunt related to a POF, as a functional factor, leads to a position-dependent transient pressure gradient through the interatrial septum and a preferential blood flow, which is the main cause of positional hypoxia. In particular, in the case of a major lung resection, the cardiovascular circulation undergoes physiological adjustments, and the vascular bed is reduced. In addition, the postsurgical shift of the mediastinum as well as the interatrial septum is another possible contributing factor of POS pathogenesis: interatrial atrial septum relocation enhances the blood flow through the atrial defect, increasing in the upright position.

In 1993 Smeenk and colleagues revised 23 cases of a symptomatic right-to-left shunt through a PFO after pulmonary resection for lung cancer, associated with postural-dependent dyspnea [12]. A particular finding from this manuscript is that the postsurgical POS is found mostly in right-sided resections. Ideally, the mediastinal shift is more pronounced in the case of right-sided resections with a subsequent greater stretching of the septum, probably due to the larger "dead space". More recently, Marini performed a survey of the literature and identified 46 cases of patients who underwent a lung resection and developed

unexplained dyspnea with hypoxemia that, in several cases, was posture-dependent [13]. The genesis of posture-dependent hypoxemia is described as a late complication of postsurgical intrathoracic anatomical remodeling in patients with a previously silent right-to-left shunt. On the contrary, in cases where the complication occurs shortly after thoracic surgery, the right-to-left shunt seems to be produced by a decrease in right ventricular compliance, following a sudden reduction of the pulmonary vascular bed or by the generation of a right-to-left pressure gradient, determined by the intra- and perioperative fluid overload. Concerning the management of those patients, the implantation of a transcatheter device using a minimally invasive approach seems to be the treatment of choice.

With regard to the clinical manifestation, POS is characterized by breathlessness and hypoxemia that occur in the orthostatism and resolve in clinostatism. Arterial blood desaturation and hypoxemia are the results of the mixing of deoxygenated venous blood and oxygenated arterial blood via a shunt. This situation can be produced by intracardiac anomalies, extracardiac alterations, and miscellaneous mechanisms. Intracardiac anomalies are the most common etiology, being detected in almost 90% of cases [14], and POS usually occurs in the setting of right-to-left cardiac communication at the interatrial level. Causes include PFO, atrial septal defect, atrial septal aneurysm, transposition of great vessels, and unroofed coronary sinus [4]. The most frequent extracardiac causes include pulmonary arterio-venous malformations, hepatopulmonary syndrome, interstitial lung disease or consolidation, and acute respiratory distress syndrome [15]. PFO is the most common cause of POS, often associated with several arterial deoxygenation syndromes, with a drop in the partial pressure of oxygen ($PO_2$) >4 mmHg or arterial oxygen saturation ($SaO_2$) >5% [16]. Right-to-left shunting has been noticed to occur in the presence of a secondary cardiac or pulmonary functional anomaly. Redirection of blood into the left atrium is proposed to happen due to the repositioning of the interatrial septum in an orthostatic position. POS can also be produced by lung resection, which provokes a reduction of pulmonary vascular bed area, resulting in increased vascular resistance and reduced right ventricular compliance [17].

The case of a patient who manifested sudden and severe dyspnea and arterial desaturation during the first attempts at the upright position immediately after the lung surgical procedure was described in this paper. Moreover, at the same time, the patient had neurological symptomatology due to a paradoxical embolism. In the literature, by contrast, platypnea-orthodeoxia syndrome is generally considered a late postoperative complication, arising about 1–3 months after lung surgery [18].

In our case of post-bilobectomy POS, in a context of near-normal right heart pressures, the proposed main mechanism is an anatomical remodeling when the patient assumes an upright position, as there is a slow shift of the mediastinum toward the operated side that facilitates right-to-left shunting: the atrial septum is dragged down by the weight of the shifted heart and the interatrial communication is brought directly in line with the inferior vena cava orifice. This anatomical modification facilitates the streaming of blood into the left atrium and the augmentation of the shunt [19]. This effect is also supported by the reduction of the vascular bed related to lung resection, with a consequent reduction in the right ventricular compliance [20]. In addition, the postoperative elevation of the right hemidiaphragm presumably contributed to the severe manifestation of symptoms, provoking a right ventricular compression [21].

The definitive management for POS in this contest is the closure of interatrial communication [22].

## 4. Conclusions

Platypnea-orthodeoxia syndrome after major lung resection is a rare cause of postoperative dyspnea and hypoxemia, which strongly impacts the patient's quality of life. POS is the result of a right-to-left shunt via interatrial communication, and mediastinal relocation and stretching of the atrial septum are among the functional elements necessary for the clinical manifestations. Physicians should always consider POS in patients with unexplained

positional dyspnea in order to identify its etiology and plan adequate treatment. In fact, postural dyspnea resulting from an intracardiac right-to-left shunt can be effectively treated by closing the PFO; nowadays the minimally invasive technique is the approach of choice with successful outcomes [23].

**Supplementary Materials:** The following supporting information can be downloaded at: https://www.mdpi.com/article/10.3390/surgeries4020018/s1, Video S1: Transoesophageal echocardiography; Video S2: 3D Transoesophageal echocardiography, Video S3: Transesophageal echocardiogram-guided Amplatzer Multifenestrated Septal Occluder.

**Author Contributions:** Conceptualization, C.C.Z.; methodology, C.C.Z., P.S. and A.L.; validation, F.M. and A.S.P.; formal analysis, A.P., L.D. and C.C.; investigation; resources, C.C.Z.; data curation, A.L.; writing—original draft preparation, C.C.Z., P.S. and A.L.; writing—review and editing, C.C.Z., P.S. and A.L.; supervision, F.M. and A.S.P. All authors have read and agreed to the published version of the manuscript.

**Funding:** This research received no external funding.

**Institutional Review Board Statement:** The study was conducted according to the guidelines of the Declaration of Helsinki, and approved by the Institutional Review Board of the University Hospital of Pisa (19211/22, 27 July 2022).

**Informed Consent Statement:** Informed consent was obtained from the patient to use CT scan images and videos, and to publish the paper.

**Data Availability Statement:** Data are contained within the article.

**Conflicts of Interest:** The authors declare no conflict of interest.

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
