# Peer review of "Platypnea-Orthodeoxia Syndrome Manifesting as an Early Complication after Lower Bilobectomy"

_2673-4095, doi:10.3390/surgeries4020018_

Round 1

Reviewer 1 Report

The introduction is too long and does not prepare the reader for the case that follows. The aim of this case report in not stated at the end of the introduction

In the case presentation, only relevant medical history and medication should be mentioned

There is no information concerning the preoperative cardiorespiratory workup and the mediastinal staging

What about the follow up (cardiological and oncological)?

The Table 1 is not really necessary

The brief literature review preceding the discussion is redundant. The literature review should be part of the discussion section

Author Response

Dear Reviewer,

Thanks to the reviewers for the thoughtful comments, which helped us in improving the quality of the manuscript.

The aim of the manuscript is to report in detail the case of a rare and underdiagnosed condition since less than 50 cases are reported in scientific literature, few of them developed after lung resection and previously never described in the immediate postoperative period.

Concerning the preoperative cardiorespiratory workup, the patient performed a global spirometry, which revealed a mild obstructive ventilation defect, with a Forced expiratory volume in one second (FEV1) of 1.70 L (105) and Tiffenau Index of 0.64.

The cardiological evaluation didn’t detect pathological conditions and ECG showed sinus rhythm and regular QT interval. Invasive mediastinal staging was performed through EBUS-TBNA, which didn’t reveal any mediastinal positive lymphadenopathy (lines 83-88)

As regards the oncological follow-up, the histological examination led to the diagnosis of invasive adenocarcinoma with a predominant solid pattern pT4N1 G3; the comprehensive molecular profiling did not reveal any mutation.  Even if according to the pathological stage the patient was admittable to adjuvant chemotherapy, the performance status after surgery did not allow the administration of systemic therapy.   

The patients underwent a total body CT scan 6 months after surgery which detected local cancer progression and multiple bone metastases. (lines 151-157)

Table 1 was inserted to expose in detail all the symptoms, procedures and events related to the hospitalization, as requested by precedent reviewers.

The literature review was modified and integrated into the discussion.

If you need any further information, please do not hesitate to contact us.

Kind regards

Reviewer 2 Report

1. Language: Moderate English changes required
2. Organisation of the manuscript: fair

3. Explanation:  specify in the abstract the kind of arterial hypertension (systemic or pulmonary?)
4. Quality of concepts: I think the issue could be interesting but it needs more details to compare the PFO in cardiovascular patient Than in pneumological patient. You have to insert right heart catheterisation, data of Qp/Qs, data of right atrium pressure volume and left side data in order to support your conclusion 
5. Statistical analysis: nothing to say
6. literary references in the discussion: didn't you thing the Platypnea-orthodeoxia syndrome manifesting the modification of right side heart-lung pressure and resistances? if no or yes explain the pathophysiological data to support your conclusion. 
7. development of the topic in the discussion: ok
8. Check conclusion message: ok
9. Take home massage: emphasise your conclusion 
10. concreteness of the work: 6/10

Table: I suggest to improve the quality of respiratory data (ETCO2, PEEE in ventilation timing, insert the data of postop S-G catheter or other forceful respiratory data. 

Author Response

Dear Reviewer, 

we apologize for the delay in replying.

Thanks to the reviewers for the thoughtful comments, which helped us in improving the quality of the manuscript.

  1. Systemic arterial hypotension (line 18)
  2. We don't have the data concerning right atrium pressure or left side data.

Even if these data would have provided us with further support, the shunt was so evident that the right catheterization was not performed, also in order to minimize the procedural time.

In this case, the massive passage of microbubbles from the right to the left atrium could already justify its closure. It would have been interesting to measure the oxygen saturation in the pre and post-atrial chambers, but blood gases were taken before and after the procedure. In addition, the general clinical condition clearly benefited from the closure.

  1. The pathophysiological data related to the modification of right side heart-lung pressure have been analyzed in the discussion (lines 189-204)

Table1:  The respiratory data required  have been added in the table1 ( Pression Support 14 cmH2O,

Peep3 cmH2O, FiO2 35%). S-G catheter has not been employed.

If you need any further information, please do not hesitate to contact us.

Kind regards

Round 2

Reviewer 1 Report

All issues are addressed sufficiently  

Reviewer 2 Report

Dear author 

I satisfied to revised manuscript. 

Topic is not too much original but it's relevant. 
There are few other publication in that field, hence it is a positive aspect. The conclusions are consistent with the evidence and arguments are presented correctly References are appropriate The tables and figures and videos are appropriate.

Regards